# Clinical Characteristics and Prognosis of Secondary Acute Lymphoblastic Leukemia in Patients with Multiple Myeloma during Long-Term Thalidomide Maintenance

**DOI:** 10.3390/jpm13030412

**Published:** 2023-02-25

**Authors:** Junru Liu, Beihui Huang, Jingli Gu, Juan Li

**Affiliations:** Department of Hematology, The First Affiliated Hospital, Sun Yat-sen University, Guangzhou 510080, China

**Keywords:** multiple myeloma, acute lymphoblastic leukemia, thalidomide, maintenance

## Abstract

**Background:** Secondary primary malignancies (SPM) have attracted increasing attention with the application of autologous hematopoietic stem cell transplantation (ASCT) and novel agents in multiple myeloma (MM). Secondary acute lymphoblastic leukemia (sALL) has rarely been reported, and the clinical characteristics and prognosis of sALL have not been described in detail. **Methods:** We retrospectively enrolled 179 consecutive newly diagnosed multiple myeloma (NDMM) patients undergoing bortezomib-based induction regimens followed by upfront ASCT and maintenance therapy from December 2006 to April 2018 in our center. **Results:** The median follow-up interval was 69.1 months, and 12 patients (6.7%) developed sALL during maintenance therapy. The median time from the diagnosis of MM to the occurrence of sALL was 51.1 (31.7–91.5) months. All sALL patients received thalidomide as maintenance therapy before the onset of sALL, and the median duration of thalidomide maintenance was 39.5 (24–74) months. The cumulative incidence of sALL was 6.6% and 11.2% at 5 and 10 years after the diagnosis of MM, respectively. All sALL patients presented with a B-cell immunophenotype accompanied by myeloid antigen expression according to flow cytometry analysis, and the BCR/ABL fusion gene was all negative. Only one patient had evidence of active MM, and the other patients were in stable status at the time of the diagnosis of sALL. The prognosis of most sALL patients was very poor, and the median overall survival time was 11.9 (1.1–51.2) months since the diagnosis of sALL. **Conclusions:** sALL should be considered for MM patients who developed unexplained persistent cytopenia while on long-term thalidomide maintenance treatment, particularly if it has been more than 3 years. With the increasing availability of new drugs for MM, thalidomide may be recommended for no more than 3 years. Sequential allogeneic hematopoietic stem cell transplantation was considered as soon as possible after achieving remission in order to achieve a longer survival.

## 1. Introduction

The application of autologous hematopoietic stem cell transplantation (ASCT) and novel agents has significantly prolonged the survival of patients with multiple myeloma (MM), and secondary primary malignancies (SPM) have attracted increasing attention [1,2,3]. Large population-based studies showed that MM patients had a 26% increased risk of developing any SPM when compared to the general population [4]. Kyle et al. [5] reported that MM patients developed secondary acute myeloid leukemia (AML) after receiving long-term melphalan treatment, and subsequent studies showed that alkylating agent exposure was associated with the occurrence of AML in the early 1970s [6,7]. With the application of new drugs in MM, the incidence of SPM in MM patients receiving lenalidomide maintenance therapy was significantly higher than that in the placebo group [8,9,10]. However, SPM mainly focused on AML and myelodysplastic syndrome (MDS), while secondary acute lymphoblastic leukemia (sALL) has only been reported in a few cases and has been described in clinical trials [11,12,13]. Two studies of lenalidomide maintenance following ASCT reported three cases and one case of sALL, respectively [9,14]. However, no details on their clonal origin were described. It is not yet clear how lenalidomide contributes to the development of sALL. The TT2 clinical trial showed a trend toward increased solid SPM risk from the start of maintenance therapy in the TT with thalidomide maintenance versus the without thalidomide arm [11]. To date, there have been no reports on the incidence of secondary ALL during thalidomide maintenance therapy. The relationship between thalidomide and SPM, especially between thalidomide and sALL, has rarely been reported. In addition, the clinical characteristics and prognosis of sALL have not been described in detail. Here we report 12 cases of sALL developing during long-term thalidomide maintenance therapy in 179 consecutive newly diagnosed MM patients who received bortezomib-based induction and upfront ASCT.

## 2. Methods

### 2.1. Patients

In this study, we retrospectively enrolled 179 consecutive newly diagnosed multiple myeloma (NDMM) patients [116 males (64.8%) and 63 females (35.2%)]. All patients were Chinese, and no other races were included. These patients had a median age of 54 (27–69) years at the diagnosis of MM and received bortezomib-based induction followed by upfront ASCT and maintenance therapy from December 2006 to April 2018 in our center. The Ethics Committee of the First Affiliated Hospital, Sun Yat-sen University, approved this study following the Declaration of Helsinki.

### 2.2. Treatment

All patients received an induction regimen containing bortezomib; 49 patients received the VD regimen (bortezomib + dexamethasone); and 130 patients received the PAD regimen (bortezomib + liposome doxorubicin + dexamethasone). A total of 158 patients received peripheral blood ASCT after stem cell mobilization with high-dose cyclophosphamide (3 g/m^2^) combined with granulocyte-colony stimulating factor (G-CSF); 18 patients received bone marrow ASCT; and 3 patients received mixed peripheral blood and bone marrow transplantation. The conditioning regimen was 200 mg/m^2^ of melphalan (MEL) and 140 mg/m^2^ for patients with renal insufficiency. A CVB (cyclophosphamide + etoposide + busulfan) conditioning regimen was also applied in this study because MEL had not been available in China for a few years. All patients received continuous maintenance with thalidomide and/or interferon-α-2b after ASCT. Lenalidomide was not routinely used as maintenance therapy in our study because of late approval in China and high cost.

### 2.3. Follow-Up

Follow-up was carried out every three months after ASCT in the first year, then every half a year after that. For patients with an unexplained abnormal blood cell count during maintenance therapy, bone marrow aspiration and flow cytometry for plasma cell antigens and leukemia-associated antigens were performed.

### 2.4. Statistical Analysis

All patients were followed until September 2021, and the median follow-up interval was 69.1 (7.4–171.3) months. The data was analyzed using SPSS 13.0 statistical software. Count data was compared using the chi-square (χ^2^) test, continuous data was analyzed using the *t*-test, and continuous data that was not normally distributed was analyzed using the Wilcoxon rank sum test. A survival analysis was performed using the Kaplan–Meier method, and overall survival (OS) was defined as the time from the diagnosis of sALL to death for any reason. *p* values less than 0.05 were considered statistically significant.

## 3. Results

### 3.1. The Basic Characteristics of MM Patients with and without sALL

Twelve patients (6.7%) developed sALL during maintenance therapy after receiving bortezomib-based induction followed by ASCT. The basic characteristics of NDMM with and without sALL are shown in Table 1. There were significant differences in female vs. male, patients with high-risk cytogenetic abnormalities (HR-CA) vs. those without HR-CA, and thalidomide vs. no thalidomide as maintenance therapy between the two groups. The median age at the time of MM diagnosis was 49.5 years, and 66.7% of patients with sALL were female. Twelve patients received thalidomide as maintenance therapy before the onset of sALL. None of the patients with sALL had HR-CA at the time of their MM diagnosis. Other factors such as induction regimens, conditioning regimens (CVB vs. Mel), transplant type, and M protein types were not statistically significant between the two groups.

### 3.2. Clinical Characteristics of Patients with MM and sALL

The median age at the time of sALL diagnosis was 55.5 (39–65) years, and the median time from the diagnosis of MM to the occurrence of sALL was 51.1 (31.7–91.5) months. The cumulative incidence of sALL was 6.6% and 11.2% at 5 and 10 years after the diagnosis of MM, respectively (Figure 1). Nine of them were treated with continuous thalidomide (200 mg/d) for maintenance therapy after ASCT. Two patients received thalidomide maintenance therapy first and then changed to lenalidomide due to side effects. One patient was treated with interferon-α-2b first, then changed to thalidomide because interferon-α-2b was not available in China. sALL developed after 39.5 (24–74) months of maintenance therapy with thalidomide. Eleven patients did not have leukemia-associated symptoms or signs at the time of the diagnosis of sALL, which was unexpectedly diagnosed on routine follow-up. Only one patient was diagnosed with sALL due to agranulocytosis accompanied by fever.

Clinical and laboratory characteristics of patients who developed sALL following thalidomide maintenance are described in Table 2. With respect to sALL, the median hemoglobin at the time of diagnosis was 105 g/L (IQR 90.5–117.75), the white blood cell (WBC) count was 2.70 × 10^9^/L (IQR 1.19–6.33), and the platelet count was 100.5 × 10^9^/L (IQR 55.75–147.25). Upon morphology review of the bone marrow, lymphoblasts accounted for 62.5% (IQR 21.3–80.3). All patients presented with a B cell immunophenotype according to flow cytometry analysis; none of them had a T cell-related immunophenotype. In addition, most of them had CD10 positivity accompanied by the expression of a myeloid antigen such as CD33 or CD15. Eleven patients tested negative for the BCR/ABL fusion gene (one case was not tested). Karyotyping was performed in six patients; only one patient had complex cytogenetics of 42, X,-Y,Del(1)(p12p31), +der (1), -10,-13,-16,-21 [1]/46,XY [2], and the other five patients had a normal karyotype. Six patients received induction therapy (vincristine + daunorubicin/idarubicin + prednisone ± pegaspargase); two of these patients subsequently underwent allogeneic hematopoietic stem cell transplantation; they were still disease-free survivors (DFS) until now. The DFS of the two patients was 19 months and 28 months, respectively. However, the other four patients died of infection during induction therapy. The median overall survival from the diagnosis of sALL was 11.9 months (range 1.1–51.2), as shown in Figure 2. With respect to MM, only one patient had evidence of active MM, and the other patients were stable at the time of the diagnosis of sALL.

### 3.3. Case Review

In this section, we will take two representative cases of sALL, as follows: Case 1, female, 32 years old, was diagnosed with MM (IgG-λ type) in December 2006. She received four cycles of induction chemotherapy with the VD regimen followed by autologous bone marrow transplantation (a conditioning regimen with melphalan 200 mg/m^2^). Thalidomide (200 mg/d) was given as maintenance therapy from January 2008. On the 74th month of thalidomide maintenance therapy (March 2014), the patient was admitted for a routine examination to evaluate the response of MM. The routine blood test showed the WBC count was 6.44 × 10^9^/L, the absolute neutrophil count (ANC) was 4.89 × 10^9^/L, Hb was 109 g/L, and the PLT count was 347 × 10^9^/L. Bone marrow smears showed 45% lymphoblasts, and flow cytometry suggested that 16.1% of CD34^+^ cells expressed HLA^−^DR^+^CD22^+^CD19^+^CD10^+^CD79a^+^CD33^+^. FISH testing was negative for the BCR/ABL fusion gene. The ALL-related prognostic gene was all negative. Because the patient did not have leukemia-related symptoms at that time and the count of blood cells was nearly normal, she chose not to receive any sALL-targeted therapy. Meanwhile, the response was evaluated as a CR for multiple myeloma. The patient was admitted again to our department in February 2016 (two years after the diagnosis of sALL) due to bone pain. Her WBC count was 3.27 × 10^9^/L, ANC was 1.69 × 10^9^/L, Hb was 57 g/L, PLT count was 70 × 10^9^/L, and a bone marrow smear showed that she was 76% lymphoblast with the same immunophenotyping results as before. She received the VILP regimen of induction chemotherapy (vincristine, idarubicin, pegaspargase, and prednisone) and obtained CR. She then received one course of IA (idarubicin + cytarabine) as consolidation chemotherapy and did not receive any further sALL-targeted treatment. The patient died of sepsis during reinduction chemotherapy with the VILP regimen because of a relapse in May 2018, 51.2 months after the diagnosis of sALL. Her myeloma remained in complete remission at that time.

Case 9, female, 44 years old, was diagnosed with MM, IgG-κ type (R-ISS stage I) in May 2017. The patient received four cycles of induction treatment with the PAD regimen and achieved very good partial remission (VGPR). After that, high-dose cyclophosphamide (3 g/m^2^) combined with granulocyte-colony stimulating factor (G-CSF) was given to mobilize hematopoietic stem cells. Then she received autologous stem cell transplantation with CVB regimen conditioning (cyclophosphamide, etoposide, and busulfan). In December 2017, thalidomide (200 mg/d) was given as maintenance therapy. On the 32nd month of thalidomide maintenance therapy, the patient was admitted for a routine examination to evaluate the response of MM. The hemogram showed a WBC count of 12.69 × 10^9^/L; an Hb of 89 g/L; and a PLT of 52 × 10^9^/L; a bone marrow smear showed 62% lymphoblasts, and the immunophenotype was CD19^+^CD10^+^CD20^−^CD22^+^CD38^+^CD34^+^HLA-DR^+^CD33^+^CD13^+^CD79a^+^. The patient was diagnosed with ALL (B-cell type); the BCR/ABL fusion gene and the ALL-related prognostic gene were negative. She received the VIP regimen of induction chemotherapy (vincristine + idarubicin + prednisone) and obtained CR. Then she received HLA-matched sibling allogeneic hematopoietic stem cell transplantation with a conditioning regimen of BuCY (busulfan + cyclophosphamide). Up to now, the patient has been disease-free and has been evaluated as CR for multiple myeloma.

## 4. Discussion

Secondary hematologic malignancies are a known risk in patients with myeloma, and they focus mainly on AML/MDS [1,15]. Originally, this finding was attributed to the application of melphalan and other alkylating agents, and more recently, the use of lenalidomide [16,17]. However, acute lymphoblastic leukemia as a second primary malignancy is rarely described in the literature. Herein, we describe a series of ALL cases secondary to MM after receiving bortezomib-based induction therapy followed by ASCT and long-term thalidomide maintenance. We also analyzed the clinical characteristics and prognosis of these patients in detail. SPM development is likely due to multiple factors. In addition to treatment-related factors, other possible risk factors may be classified as either disease-related or host-related, such as age, sex, and genetics [3,15]. Our study showed that the incidence of sALL was higher in female patients than in male patients, which differed from SEER database data that reported a higher proportion of male patients [18]. This may be related to the different tumor types of SPM. None of the patients with sALL expressed high-risk cytogenetics at the time of their MM diagnosis. According to Engelhardt et al. [19], patients without high-risk cytogenetics have a higher chance of long-term survival and a longer maintenance period, which may lead to an increase in the incidence of SPM. 

In our study, univariate analysis showed that thalidomide vs. no thalidomide made a significant difference in the incidence of sALL. However, multivariate analysis was not significant. This may be due to the small number of cases and the insufficient time for follow-up. We will continue to follow up on these patients to confirm the significance of thalidomide in the occurrence of sALL. Whether thalidomide causes SPM and sALL, as well as the possible underlying mechanism, remains unclear. Since both thalidomide and lenalidomide are immunomodulators, it is possible that thalidomide induces SPM and sALL through a mechanism similar to that of lenalidomide. Studies have found that immunosuppressive therapy can reactivate the EBV lytic cycle of quiescent B lymphocytes and lead to immunosuppression [20]. Another potential effect of lenalidomide on B cells is the alteration of the CRL4 E3 ubiquitin ligase complex through the cereblon protein, i.e., the selective ubiquitination and degradation of the IKAROS protein (encoded by IKZF1) and AIOLOS protein (encoded by IKZF3) [21,22]. IKAROS and AIOLOS transcription factors are critical for the regulation of B-cell function. Lenalidomide can also amplify regulatory T cells (Tregs) to induce immune tolerance [23]. Three cases of sALL during lenalidomide maintenance therapy were analyzed, and sALL usually occurred after long-term lenalidomide maintenance therapy (33–92 months), so the occurrence of sALL may be time-dependent rather than dose-dependent [24]. In the 12 patients reported in our study, sALL occurred during long-term thalidomide maintenance therapy, and the median duration was 39.5 months. Was the sALL caused by thalidomide also time-dependent rather than dose-dependent? Whether any of these effects of thalidomide play a role in the development of B-cell sALL in MM patients is worthy of further study.

Both MM and B-ALL belong to B-cell-derived malignancies, and their sequential development in the same patient may suggest a MM clonal dedifferentiation into a more aggressive form of B-cell malignancy, such as B-cell acute leukemia. Exome sequencing analysis of paired DNA samples from MM and sALL illustrated that the neoplasms were clonally unrelated, contradicting this hypothesis [25]. Ueda reported the occurrence of sALL in one MM patient undergoing alkylating agent therapy, and this patient exhibited MLL abnormalities [26]. Further IgH rearrangements confirmed that MM and sALL came from different lymphocyte clones, suggesting that the modification of the MLL gene resulted in the development of sALL during tumorigenesis. However, the study by Lau et al. showed one case of MM developing sALL three years after transplantation did not have MLL gene abnormalities, and two independent B cell populations were involved in the formation of two types of lymphocyte tumors at different time points [27]. All 12 cases in our study had a B-lymphocyte immunophenotype; there were no T-lymphocyte-related markers, which was similar to what has been reported in other published articles [24,25,26,27,28]. Most of the sALL tend to have some high-risk cytogenetics and molecular features, such as 11q23 rearrangement, as well as BCR-ABL, and complex chromosomal abnormalities [29]. However, karyotype analysis was performed in six patients; only one patient had complex cytogenetics, and the other five patients had a normal karyotype. Eleven patients tested negative for the BCR/ABL fusion gene (one case was not tested). In addition, in all cases, the ALL-related prognostic gene was negative. It was not clear whether MM and sALL originated from the same B-cell clone because the IgH gene rearrangement test was not performed in our study. As shown in this study, the immunophenotypes of sALL patients were mostly accompanied by the weak expression of the myeloid antigens CD33 or CD15, which was not reported in a previous study. Therefore, we think that these sALL patients may have treatment-related leukemia as the result of MM therapy before presenting with a B-lymphocytic cell phenotype rather than the more frequently seen myeloid phenotype.

The clinical characteristics of 12 cases of sALL were also analyzed. The protein type was IgG or IgA in ten of the cases, with only two being light chain types. Most of them had no unique clinical presentation of sALL and were diagnosed on routine follow-up of MM. It was worth noting that the majority of them had cytopenia that was thought to be related to thalidomide maintenance effects. sALL should be suspected in MM patients who develop unexplained persistent cytopenia. The prognosis for most of the patients was very poor, and four of them died of infection during induction therapy. Two patients who underwent allogeneic hematopoietic stem cell transplantation have been disease-free until now. However, we also noted that one patient (case 1) remained stable without receiving any sALL-targeted treatment within two years after the diagnosis of sALL. Khan et al. described two cases of B-cell acute lymphocytic leukemia with a unique presentation who received lenalidomide, generally with good prognostic features and a good response to standard chemotherapy [13]. The clinical and biological characteristics and prognosis of ALL secondary to MM in this special population need to be further studied. This paper also suggests that the possibility of secondary ALL should be considered for MM patients who have received long-term thalidomide maintenance treatment once abnormal blood cell counts and blast cells are found in bone marrow smears. With the increasing availability of new drugs for MM, thalidomide may be recommended for no more than 3 years. In order to achieve a longer survival, these sALL patients were considered for allogeneic stem cell transplant as soon as possible after remission.

## Figures and Tables

**Figure 1 jpm-13-00412-f001:**
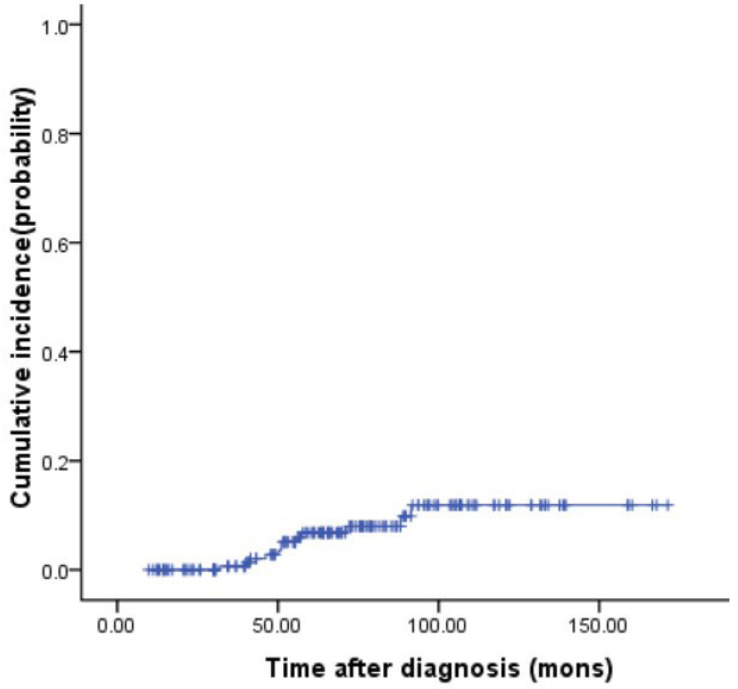
Cumulative incidence rates of sALL in 179 multiple myeloma patients.

**Figure 2 jpm-13-00412-f002:**
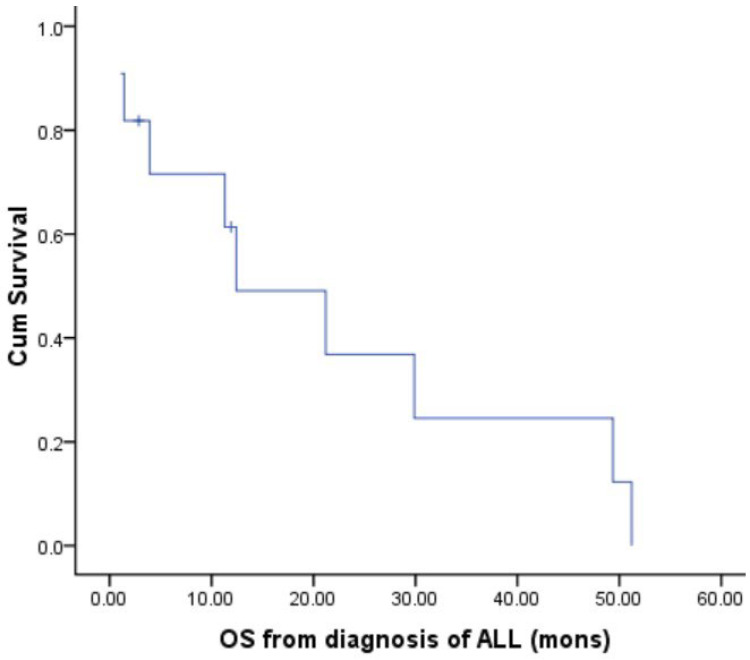
OS from the diagnosis of sALL.

**Table 1 jpm-13-00412-t001:** The basic characteristics of patients with MM with sALL and without sALL [n (%)].

Factor	with sALL (n = 12)	without sALL (n = 167)	*p* Value
**Gender (M/F)**	4 (33.3)/8 (66.7)	112 (67.1)/55 (32.9)	0.022
**Median age (years)**	49.5 (32–60)	54 (27–69)	0.975
**M protein type**			0.825
IgG/IgA/LCO/IgD	8 (66.7%)/2 (16.7%)/2 (16.7%)/0	91 (54.5%)/33 (19.8%)/38 (22.8%)/5 (3.0%)	
**R-ISS stage I/II/III**	4 (33.3)/6 (50.0)/2 (16.7)	41 (27.0)/92 (60.5)/19 (12.5)	0.771
**Plasma cells (%)**	20.3 (7–68)	23.0 (0.5–94.5)	0.599
**Hb (≤100 g/L)**	8/12 (66.7)	93/167 (55.7)	0.459
**Renal insufficiency**	4/12 (33.3)	25/167 (15.0)	0.095
**Hypercalcemia (>2.75 mmol/L)**	1/12 (8.3%)	22/167 (13.2%)	0.628
**PLT ≤ 100 × 10^9^/L**	1/12 (8.3%)	10/163 (6.1%)	0.762
**LDH (≥240 U/L)**	3/12 (25.0)	22/158 (13.9)	0.296
**HR-CA ***	0/9	33/107 (30.8)	0.049
**Induction therapy**			0.117
VD/PAD	1 (8.3)/11 (91.7)	49 (29.3%)/118 (70.7)	
**Transplantation type**			
PBSC/BM ± PBSC	9 (75.0)/3 (25.0)	149 (89.2)/18 (10.8)	0.139
**Conditioning regimen**			
MEL/CVB	7 (58.3%)/5 (41.7)	90 (53.9)/77 (46.1)	0.766
**Maintenance therapy**			0.036
Thal/non-Thal	12 (100)/0	119 (74.8)/40 (23.4)	

* HR-CA: any t(4;14), t(14;16), or del(17p) detected by FISH.

**Table 2 jpm-13-00412-t002:** Characteristics and prognosis of MM patients who developed sALL.

Patient	Sex	Age *	M-Type	Thal Duration (mons)	Age ^#^	Treatment	Survival State and Cause of Death	OS from sALL (mons)
Case 1	F	32	IgG-λ	74	39	VILP induction and IA consolidation, then relapse and VILP reinduction	Death/Infection during reinduction therapy	51.2
Case 2	F	49	IgA-κ	24	52	No treatment	Death/Infection	21.2
Case 3	M	60	IgG-κ	31	63	Unknown chemotherapy	Death/Not available	11.3
Case 4	F	60	IgG-λ	38	64	VILP induction	Death/Infection during induction	29.9
Case 5	F	47	κ	41	51	No treatment	Death/Not available	12.4
Case 6	F	59	IgG-λ	52	64	VILP induction	Death/Infection during induction	1.37
Case7	F	51	IgG-κ	50	56	VDLP induction	Death/Infection during induction	1.10
Case 8	M	59	IgG-λ	38	64	No treatment	Death/Not available	3.03
Case 9	F	44	IgG-κ	32	48	VIP induction followed by ALLO-HSCT	Alive	/
Case 10	M	59	IgG-λ	46	65	Unknown chemotherapy	Death/Not available	1.43
Case 11	F	47	κ	36	55	VDLP induction followed by ALLO-HSCT	Alive	/
Case 12	M	39	IgA-κ	44	44	Newly diagnosed	Not evaluated	Not evaluated

* age of diagnosis with MM, ^#^ age of diagnosis with sALL.

## Data Availability

Data is unavailable due to ethical restrictions.

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
