# Peer review of "Clinical Characteristics and Prognosis of Secondary Acute Lymphoblastic Leukemia in Patients with Multiple Myeloma during Long-Term Thalidomide Maintenance"

_jpm, 2023, doi:10.3390/jpm13030412_

Round 1
Reviewer 1 Report
The paper entitled: “Clinical characteristics and prognosis of secondary acute lymphoblastic leukemia in patients with multiple myeloma during long-term thalidomide maintenance” (jpm-2204824) by Junru Liu et al. reports the clinical characteristics and outcome of sALL in MM patients after thal maintenance.
Albeit the paper is well written, prepared and of special interest, comments should be addressed.
Comments:
1. Table 1: the authors should analyze Mel vs. CVB and thal vs. no thal separately according their significance.
2. Additionally, it is unclear whether the effect of thal maintenance is the only cause for developing sALL. The authors should provide further univariable and multivariable analyses to underline the hypothesis.
3. Conclusions: the authors should focus on 1. What can be learnt for clinical practice, 2. What are the exact recommendations for the clinicians?
4. Page 2, 8 and 9: the authors should add more references after their statements.
Author Response
- Table 1: the authors should analyze Mel vs. CVB and thal vs. no thal separately according their significance.
Response:Thank you for your advice. We analyzed the effect of Mel vs. CVB and thal vs. no thal separately. There was no significant difference between Mel and CVB on the occurrence of sALL. However, the incidence of sALL was higher in patients with thal than that in patients without thalidomide, which was statistically different between the two groups, as shown in Table 1.
- Additionally, it is unclear whether the effect of thal maintenance is the only cause for developing sALL. The authors should provide further univariable and multivariable analyses to underline the hypothesis.
Response:Thank you for your advice. Univariate analysis showed that thal vs. no thal has significant difference for the incidence of sALL. However, multivariate analysis was not significant. We think that this may be related to the small number of cases and the time of follow-up was not enough. We will continue to observe these patients to further study the significance of thalidomide in the occurrence of sALL.
- Conclusions: the authors should focus on 1. What can be learnt for clinical practice, 2. What are the exact recommendations for the clinicians?
Response:Thank you for your advice. It has been revised in the section of conclusion based on your comments. sALL should be considered for MM patients who developed mysterious persistent cytopenia during long-term thalidomide maintenance treatment especially for more than 3 years. With the increasing availability of new drugs for MM, thalidomide may be recommended for no more than 3 years. Sequential allogeneic hematopoietic stem cell transplantation was considered as soon as possible after achieving remission in order to achieve a longer survival.
- Page 2, 8 and 9: the authors should add more references after their statements.
Response:More references have been added after their statements in the section of introduction and discussion.
Reviewer 2 Report
Thalidomide is not a standard practice anymore in most countries. Despite this, the paper sheds light on a rare but important long term complication of IMiDs. This may alert readers to look for sALL in patients on long term lenalidomide which is widely used in MM patients. It is overall written well and I do not have any major suggestions for revisions. It would have been interesting to look at the next generation sequencing of those patients and to note if most had alterations in IKZF1. If this is available, it may be worth including.
Also, please include information about karyotypes/cytogenetics of the sALL. Did any of those patients had complex cytogenetics?
Author Response
Also, please include information about karyotypes/cytogenetics of the sALL. Did any of those patients had complex cytogenetics?
Response:Thank you for your suggestion. The information of karyotypes has been added in the result section. Karyotype was performed in six patients, only one patient had complex cytogenetics, and the other five patients had normal karyotype. Unfortunately, we did not perform next-generation sequencing. We will perform next-generation sequencing once available in the future.
Reviewer 3 Report
The authors described their institutional experience with secondary ALL. They reported higher than expected cumulative sALL of 11.5 % at 10 years. The authors suggested this could be due to thalidomide maintenance, but further studies are needed.
Here are a few points to help improve this work:
1. In the abstract and conclusion section: the authors recommended an allogenic transplant based on two patients. I would suggest toning down this recommendation and making a "consideration for allogeneic stem cell transplant."
2. Instead of mentioning that the patients are still disease-free until now. Please, mention how many months that is.
3. What was the re-induction regimen for case 1?
4. Kindly go over typos in the manuscript.
5. Curious to know, are those all Chinese (Asian) patients? what there any other races included in the study? This information might be worth adding because of the high incidence of sALL.
Author Response
Here are a few points to help improve this work:
- In the abstract and conclusion section: the authors recommended an allogenic transplant based on two patients. I would suggest toning down this recommendation and making a "consideration for allogeneic stem cell transplant."
Response:Thank you for your advice. We agree with your suggestion and have revised it.
- Instead of mentioning that the patients are still disease-free until now. Please, mention how many months that is.
Response:I apologize for not indicating the timing of disease -free. The DFS of the two patients were 19 months and 28 months, respectively. It has been revised in the paper.
- What was the re-induction regimen for case 1?
Response: The re-induction regimen of Case 1 was VILP regimen, the same as the first reduction. It has been included in the paper.
- Kindly go over typos in the manuscript.
Response:Thank you for your suggestion. We have checked and corrected word by word.
- Curious to know, are those all Chinese (Asian) patients? what there any other races included in the study? This information might be worth adding because of the high incidence of sALL.
Response: All patients were Chinese, and no other races were included. We have included this information in the methods section.
Round 2
Reviewer 1 Report
The paper entitled: “Clinical characteristics and prognosis of secondary acute lymphoblastic leukemia in patients with multiple myeloma during long-term thalidomide maintenance” (jpm-2204824) by Junru Liu et al. reports the clinical characteristics and outcome of sALL in MM patients after thal maintenance.
The authors addressed all my initial comments adequately